# Iris Racemose Hemangioma Assessment with Swept Source Optical Coherence Tomography Angiography: A Feasibility Study and Stand-Alone Comparison

**DOI:** 10.3390/jcm11216575

**Published:** 2022-11-06

**Authors:** Filippo Confalonieri, Huy Bao Ngo, Helga Halldorsdottir Petersen, Nils Andreas Eide, Goran Petrovski

**Affiliations:** 1Department of Ophthalmology, Oslo University Hospital, Kirkeveien 166, 0450 Oslo, Norway; 2Center for Eye Research and Innovative Diagnostics, Department of Ophthalmology, Institute for Clinical Medicine, University of Oslo, Kirkeveien 166, 0450 Oslo, Norway; 3Department of Biomedical Sciences, Humanitas University, Via Rita Levi Montalcini 4, Pieve Emanuele, 20090 Milan, Italy; 4Department of Ophthalmology, University of Split School of Medicine and University Hospital Centre, 21000 Split, Croatia

**Keywords:** iris racemose hemangioma, iris arteriovenous malformation, iris arteriovenous aneurysm, Swept Source Optical Coherence Tomography (SS-OCT), OCT angiography (OCTA)

## Abstract

Purpose: To evaluate arteriovenous malformations (AVM) with swept-source (SS) optical coherence tomography (OCT) angiography (OCTA) in iris racemose hemangioma and compare it with traditional intravenous iris fluorescein angiography (IVFA). Methods: A cross-sectional observational clinical study was conducted on patients with iris racemose hemangioma with the ZEISS PLEX Elite 9000 SS OCT & OCTA. Results: Three eyes of three patients were imaged. Iris racemose hemangiomas demonstrated a tortuous, well-defined, and continuous course of the AVM. The ZEISS PLEX Elite 9000 SS OCT & OCTA allowed for a detailed visualization of the ARM and was superior to IVFA in depicting small caliber, fine vessels. Conclusions: SS-OCTA may provide a dye-free, no-injection, cost-effective method comparable to spectral domain OCTA and IVFA for diagnosing and monitoring iris racemose hemangiomas for growth and vascularity.

## 1. Introduction

The iris racemose hemangioma, also known as iris arteriovenous malformation (AVM) and AV aneurysm, is a benign vascular lesion [1]. Its fundamental pathophysiology is an artery that joins directly to a vein, bypassing the capillary network. A true AVM of the iris is rare, and little has been published about this entity [2].

Traditionally, intravenous iris fluorescein angiography (IVFA) has been regarded as the diagnostic procedure of choice for iris racemose hemangioma. IVFA shows a hyperfluorescent lesion that is rapidly filling in, and shown nearly no mid-to-late phase leaking. Additionally, intervening iris hypoperfusion can be seen. Iris indocyanine green angiography can be utilized to better penetrate through the iris pigmentation in individuals with darker irides [2,3].

Spectral domain (SD) optical coherence tomography angiography (OCTA) has previously been shown to be a useful tool in physiologic iris vascular imaging [4]. Iris racemose hemangioma has previously been shown to have distinct features on SD-OCTA only in one descriptive, noncomparative case series [5].

The ZEISS PLEX Elite 9000 Swept-Source (SS) OCT & OCTA is an evolution of the previous SD-OCTA technology [6] which generates high-resolution three-dimensional maps of the retinal and choroidal microvasculature. SS-OCTA has previously been established as an effective, non-invasive tool in the management of corneal and iris neovascularization [7,8]. To our knowledge, no validation report exists on SS-OCTA capability to image iris racemose hemangioma, which we hereby report and discuss the results of by comparing them to traditional IVFA imaging capabilities.

## 2. Materials and Methods

This is a retrospective, comparative, observational case series of 3 consecutive unilateral iris racemose hemangiomas seen at the Department of Ophthalmology of the Oslo University Hospital (OUH), Oslo, Norway from October 2020 to February 2022. Institutional review board approval from the OUH was not necessary, but written consent was obtained from all patients and approval from the data protection officer at OUH were obtained.

All patients underwent comprehensive ophthalmic examination including slit lamp evaluation with gonioscopy, and anterior segment (AS) photography, dilated funduscopic examination, ultrasound biomicroscopy (UBM), posterior segment OCT, AS OCT, IVFA and OCTA. The latter was performed using a ZEISS PLEX Elite 9000 SS OCT & OCTA, (Carl Zeiss Meditec AG Goeschwitzer Str. 51-52 07745 Jena, Germany) [6]. For AS SS-OCTA, a volume cube scan protocol was used to scan the lesion (3 × 3 mm; AngioRetina); however, the positioning and focus settings were adjusted manually to allow visualization of the AS, and to obtain a precise focus of the OCT B-scan.

Patient demographics (age, ethnicity, and sex) were recorded. Data regarding clinical features included the affected eye, best corrected visual acuity (BCVA), iris color, iris racemose hemangioma location, type, and size (number of clock hours), presence of iris or ciliary body mass, and other concomitant ocular lesions. Imaging features were recorded from AS photography, AS SS-OCT, SS-OCT, OCTA, IVFA, and ocular ultrasound. The imaging results of the vascular course analysis obtained with AS SS-OCT were then qualitatively compared to that obtained with IVFA.

## 3. Results

The mean age at the first referral was 62.6 years (range, 59–69 years), and all patients were Caucasian with measured BCVA in the involved eye equal to 20/20 (n = 2) and 20/32 (n = 1). The patients were both male (n = 2) and female (n = 1) and they were all referred with the diagnosis of a pigmented iris lesion. The eyes studied were both right (n = 2) and left (n = 1). The iris color was blue in all cases and no concomitant iris or ciliary body masses were found. Both at the slit lamp and at the UBM imaging the lesions appeared structurally ill defined. Specifically, by slit lamp biomicroscopy, the iris racemose hemangiomas were barely noticeable within the iris stroma as a relatively long, dark-red vessel with a tortuous course, buried within the normal stromal tissue without solid tumor in the iris or ciliary body. The hemangioma was located temporally in the right (n = 2) or nasal in the left (n = 1) eye and demonstrated up to 4 clock hours of involvement. AS-OCT showed cystic lesions in the iris stroma in all cases, which likely correspond to the large vascular lumen. All lesions were classified as complex type due to intertwining convolutions, and not a simple loop. Table 1 summarizes the features of the lesions.

### 3.1. Patient 1

A 59-year-old male was referred for increased intraocular pressure (IOP) in the left eye and an asymptomatic abnormal iris pigmentation in the right eye. The iris lesion in the right eye was diagnosed as complex iris racemose hemangioma; no treatment was required, and the patient was followed-up. Figure 1 shows the multimodal imaging pertinent to the iris lesion. The structural AS-OCT shows hyporeflective cystic cavities representing the lumen of the racemose hemangioma of the iris. These structures appear localized in the anterior iris stroma and have variable dimensions determined by the direction of the vessels in relation to the cross section of the AS-OCT. Figure 2 compares the late phase IVFA with the iris OCTA, showing that the convoluted vascular network from 7 to 8 o’clock is equally discernible with both of the imaging techniques and nearly no leakage is present at IVFA.

### 3.2. Patient 2

A 69-year-old female was referred for herpes zoster ophthalmicus in the area of the right ophthalmic branch of the trigeminal nerve. Incidentally, after the resolution of the eczematous lesion, dry age-related macular degeneration was diagnosed and an abnormal, asymptomatic iris pigmentation in the right eye was noted. The iris lesion in the right eye was diagnosed as complex iris racemose hemangioma, no treatment was required, and the patient was followed-up. Figure 3 shows the multimodal imaging pertinent to the iris lesion. The racemose hemangioma of the iris is shown as hyporeflective cystic cavities on the structural AS-OCT. These structures appear deep in the iris stroma. Since the structural AS-OCT scan is acquired in correspondence of a horizontal running, relatively non-convoluted large vessel, these cystic spaces look quite different than the corresponding image in the previous case. Figure 4 compares the late phase IVFA with the iris OCTA, showing that the convoluted vascular network from 6 to 10 o’clock is equally discernible with both the imaging techniques and nearly no leakage is present at IVFA.

### 3.3. Patient 3

A 60-year-old male was referred for an asymptomatic iris pigmented lesion in the left eye noted right after bilateral refractive lens exchange surgery. The iris lesion in the left eye was diagnosed as complex iris racemose hemangioma, no treatment was required, and the patient was followed-up. The structural AS-OCT shows two hyporeflective cystic cavities representing the lumen of the racemose hemangioma of the iris forming a loop. The septum between the two cavities represents the vessel wall that folds back in on itself. The slab represents the area of OCTA flow acquisition, and it shows no specific signal inside the cystic cavities, probably due to the disruption of laminar flow inside the vascular malformation. Figure 5 shows the multimodal imaging pertinent to the iris lesion. Figure 6 compares the late phase IVFA with the iris OCTA, showing that the convoluted vascular network from 6 to 10 o’clock is equally discernible with both the imaging techniques and nearly no leakage is present at IVFA.

All the iris racemose hemangiomas showed a clear structural resolution by SS-OCTA imaging, comparable to the resolution obtained with IVFA. The fine caliber vessels of the irides studied were better discernible by SS-OCTA imaging than by IVFA.

No associated vascular malformation was detected on magnetic resonance imaging (MRI) of the head and orbits in the three patients studied.

## 4. Discussion

AS SD-OCTA has been demonstrated to be a useful and reliable tool for the vascular imaging of both normal irides and benign or malignant lesions of the iris, with possible applications in vascular conjunctival lesions, as well [4,5,9,10,11].

Using AS SD-OCTA, Skalet et al. found that iris melanomas contain higher intertumoral vascular density than benign iris nevi [9]. Chien et al. employed SD-OCTA to identify extensive vasculature within conjunctival racemose hemangioma, which was not visible on FA [11]. Kang et al. showed iris microhemangiomatosis by SD-OCTA, outlining small vascular lesions with flow signal from the posterior iris stroma [10]. Furthermore, Chien et al. [5] showed that SD-OCTA clearly depicts the looping course of iris racemose hemangiomas and can highlight the fine details of radial iris vessels, indistinguishable by IVFA [5]. The same result of superiority in the resolution of the vessels by SD-OCTA over IVFA was described by Zett et al. in normal iris [12].

SS-OCTA has recently emerged as an imaging technology, and it has been described to be able to better define the iris vessels in normal pigmented iris than SD-OCTA [13].

To our knowledge, this is the first study to report the outcome of racemic iris hemangiomas when imaged with SS-OCTA.

On SS-OCTA imaging, all our studied iris racemose hemangiomas displayed a sharp delineation, comparable to the resolution attained with IVFA in large caliber vessels and superior in small caliber vessels. By using SS-OCTA, the normal fine caliber vessels of the irides studied could also be more clearly delineated than by IVFA, which is consistent with the previous findings conducted with SD-OCTA [5]. These findings suggest that OCTA in general, and possibly SS-OCTA more than SD-OCTA, might be able to detect smaller caliber, abnormal vessels when they are not yet detectable on slit lamp examination or by IVFA. The application of SS-OCTA can find wider applications in other neovascular diseases involving the iris, such as that of early imaging of iris neovascularization in neovascular glaucoma—a pathologic process that has just been recently investigated for the first time using SS-OCTA [14].

The patients enrolled in this study had lightly pigmented irides. The iris color affects the ability to detect flow signals by OCTA, and flow is most effectively detected in irides that are lightly pigmented [9]. This means that the same resolution of the vascular lesions might not be attainable in darkly pigmented irides, even though the higher transmittance of the signal and higher resolution found with fine vessels in SS-OCTA imaging suggests that this technique would anyway be capable of better image iris vessels compared to IVFA.

Even though this study aims at highlighting the potential of the recently introduced SS-OCTA technology, it is crucial to understand that each strategy has a number of drawbacks. The light penetrance of deeper tissue and vascular leakage, which can hide microvascular features, restricts the use of IVFA. On the other hand, the limitations of SS-OCTA and OCTA technology in general, are due to the motion artifacts and its inability to recognize specific flow patterns, in particular, vascular leakage. Since iris racemose hemangiomas do not leak at IVFA, as opposed to iris neovascularization, which demonstrates a specifically modest dye leakage, IVFA may be still superior to differentiate iris vascular diseases in circumstances when they are difficult to distinguish. On the contrary, the fine, clinically invisible iris vessels can be detected by OCTA in the very early stages, as well as in the regressed stage of iris neovascularization [15]. In addition, leakage IVFA may be superior in differentiating iris racemose hemangioma from other vascular or solid tumors of the iris that display a pathologic vessel wall structure, such as malignant neoplasms [16].

The main drawback of our research is the lack of direct comparison between the information derived from SS-OCTA and SD-OCTA to determine if there is an actual clinical advantage. Further studies are required to determine the clinical advantage of SS-OCTA in iris vascular imaging over other imaging techniques.

## 5. Conclusions

SS-OCTA may provide a dye-free, no-injection, cost-effective method that is comparable and complementary to SD-OCTA and IVFA for diagnosing and monitoring iris racemose hemangiomas for growth and vascularity.

## Figures and Tables

**Figure 1 jcm-11-06575-f001:**
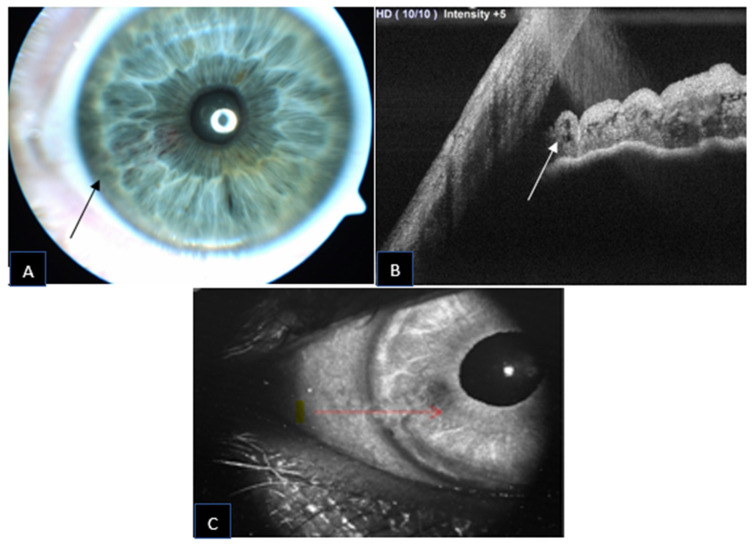
Iris racemose hemangioma appearance on slit lamp photography ((**A**); black arrow) with a corresponding AS-OCT ((**B**); white arrow) and infrared photograph ((**C**); red arrow). The arrows indicate the inferior border of the lesion.

**Figure 2 jcm-11-06575-f002:**
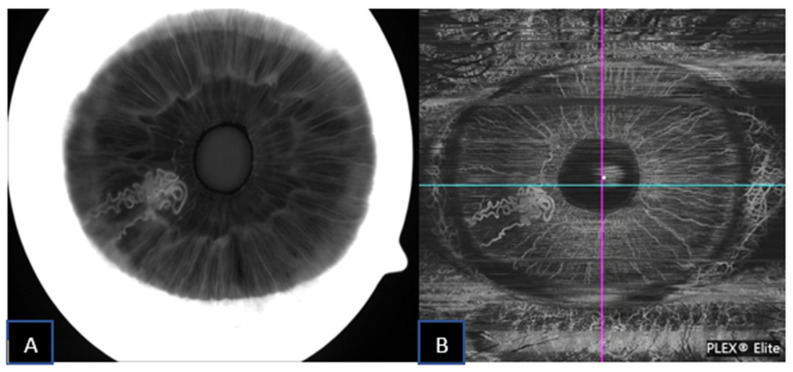
Iris racemose hemangioma under late-phase IVFA (**A**) and corresponding SS-OCTA (**B**) imaging.

**Figure 3 jcm-11-06575-f003:**
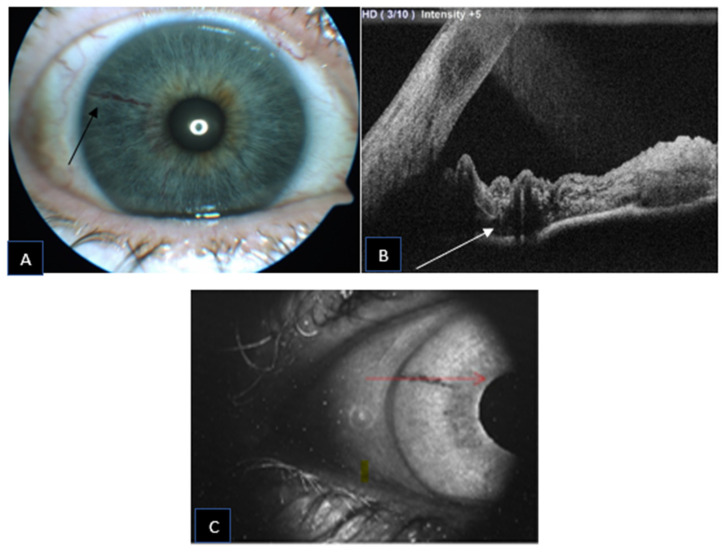
Iris racemose hemangioma appearance on slit lamp photography ((**A**); black arrow) with a corresponding AS-OCT ((**B**); white arrow) and infrared photograph ((**C**); red arrow). The arrows indicate the superior border of the lesion.

**Figure 4 jcm-11-06575-f004:**
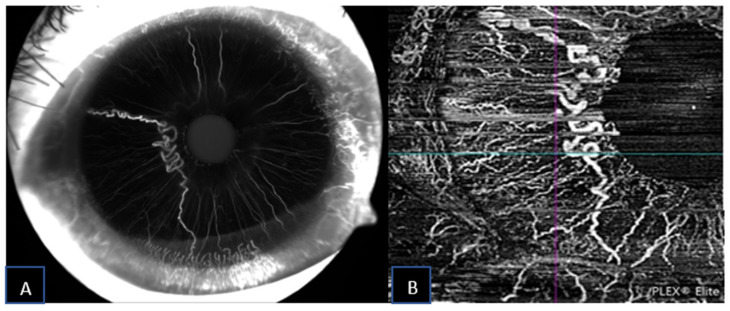
Iris racemose hemangioma under late-phase IVFA (**A**) and corresponding SS-OCTA (**B**) imaging.

**Figure 5 jcm-11-06575-f005:**
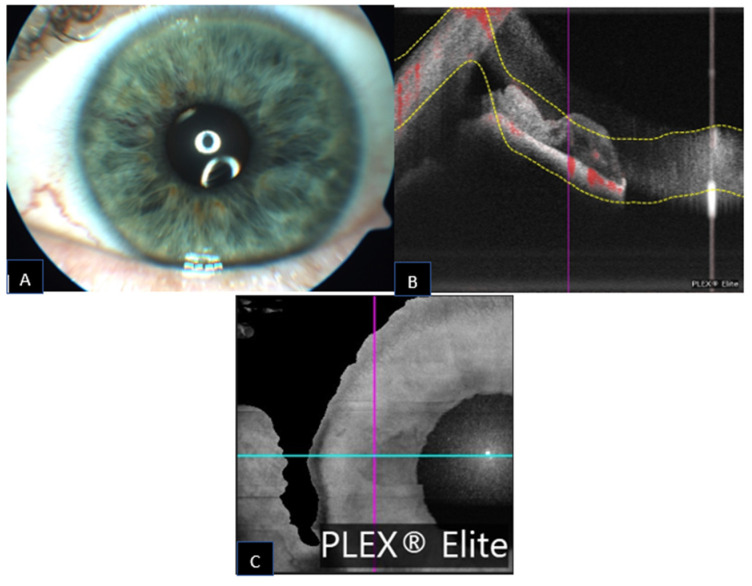
Iris racemose hemangioma appearance on slit lamp photography (**A**) and a corresponding AS-OCT with superimposed flow signal (**B**) and infrared photograph (**C**) with the blue line corresponding to the cross-sectional OCT image plane acquisition.

**Figure 6 jcm-11-06575-f006:**
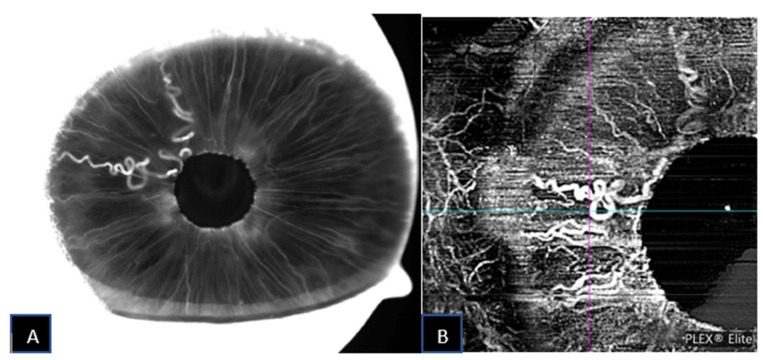
Iris racemose hemangioma under late-phase IVFA (**A**) and corresponding SS-OCTA (**B**) imaging.

**Table 1 jcm-11-06575-t001:** The demographics, clinical and SS-OCTA imaging features of the patients having iris racemose hemangioma.

Demographics	Patient 1	Patient 2	Patient 3
**Age**	59	69	60
**Ethnicity**	Caucasian	Caucasian	Caucasian
**Sex**	Male	Female	Male
**Referral diagnosis**	Pigmented iris lesion	Pigmented iris lesion	Pigmented iris lesion
**Involved eye**	Right eye	Right eye	Left eye
**BCVA**	20/20	20/32	20/20
**Iris color**	Blue	Blue	Blue
**Clock hour**	7–8	6–10	10–12
**Concomitant iris or ciliary body mass**	no	no	no
**Visibility UBM**	Indistinct	Indistinct	Indistinct
**Visibility at the slit lamp**	Red lesion	Red indistinct lesion	Red indistinct lesion
**Visibility at AS-OCT**	Cystic lesions	Cystic lesions	Cystic lesions
**Visibility at SS-OCTA**	Well defined vascular network	Well defined vascular network	Well defined vascular network
**Visibility at IVFA**	Well defined vascular network without leakage	Well defined vascular network without leakage	Well defined vascular network without leakage
**Vascular course**	Convoluted	Convoluted at the pupil margin	Convoluted at the pupil margin
**Hemangioma type characterization**	Complex	Complex	Complex

## Data Availability

Not applicable.

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
