# Peer review of "Iris Racemose Hemangioma Assessment with Swept Source Optical Coherence Tomography Angiography: A Feasibility Study and Stand-Alone Comparison"

_jcm, 2022, doi:10.3390/jcm11216575_

Round 1
Reviewer 1 Report
This is an interesting case-series including iris hemangiomas imaged by SS-OCTA.
All cases were well-presented with good-quality images.
The authors compared the OCTA and IVFA in their cases and reported the drawbacks of both techniques.
There is only one concern; please include the comparison between a SD-OCTA and SS--OCTA in such cases and discuss it.
I agree that a multimodal imaging containing OCTA in cases with iris vascular abnormalities can provide better diagnostic features.
Author Response
Dear Reviewer,
Thank you for your kind comments. We agree that SD-OCTA/SS-OCTA direct comparison would bring a major contribution to the Paper. Unfortunately, we do not have at our disposal such images and the direct comparison cannot be described here. Nevertheless, a few lines have been added at the very end of the discussion referring to this drawback in methodology and calling for the direct comparison in following papers as you correctly pointed out.
With that, we hope the Reviewer will be satisfied with our answers and make a final positive decision.
Reviewer 2 Report
Confalonieri and colleagues present a well-documented case series of 3 patients with Iris Racemose Hemangioma imaged with SS OCTA. The OCTA images are impressive and reach the quality of iris fluorescein angiography.
My comments:
Lines 39-42 should be rephrased to improve reading experience: “has been previously described”… “has previously been shown”…
Line 45 – Please add some background on use of SS OCTA in anterior segment imaging.
Table 1 – as the majority of parameters are identical between the 3 patients, this table does not add information, and could be summarized in text.
The infrared photographs do not seem to add value, it is hard to distinguish the lesion and the images appear blurry.
Figure 5B and 5C – the quality of the B-Scan is suboptimal and there are image artifacts. Please remove the slabs, as they are inaccurately posed.
The description of the structural anterior segment OCT images should be extended and include a detailed evaluation of the findings.
Author Response
Dear Reviewer,
Lines 39-42 have been rephrased to improve reading experience.
Line 45 – Some background on the use of SS-OCTA in anterior segment imaging has been added.
Table 1 has been summarized in the text. The table is also available for the reader, as we believe, it can improve the reading experience.
As per the infrared images, their main purpose is to add a localization and navigation detail for the cross-sectional OCT scan, making it possible to understand where exactly the cross-sectional OCT was acquired. The fact that the lesion is not visible also could add value by specifying that the vascular malformations are subtle. These images are also a reference for demonstrating the imaging device used and showcasing multimodal imaging techniques.
Figure 5 has been improved in the description, making the presence of the slab crucial for improving the reader’s understanding of the flow signal on OCTA. Please note that the image C is taken from another device than the other C corresponding images. Here we have the structural OCT plus the flow signal derived from the OCTA, which also correspond to En face in image 6. That is because no corresponding device image was available for patient 3, but the slab gives more information as to where the device is studying the flow. The overall number of images is then important as it allows anyway to locate where the cross-sectional OCT was acquired, and this is why they are all required.
Thank you for pointing out these aspects as they made us understand where the explanation of figures needed improvement for perfecting the reading experience.
The description of the structural anterior segment OCT images has been extended and a detailed evaluation of the findings has been included.
With that, we hope the Reviewer will be satisfied with our answers and make a final positive decision.